# Parameter Study on Force Curves of Assembled Electronic Components on Foils during Injection Overmolding Using Simulation

**DOI:** 10.3390/mi14040876

**Published:** 2023-04-19

**Authors:** Martin Hubmann, Mona Bakr, Jonas Groten, Martin Pletz, Jan Vanfleteren, Frederick Bossuyt, Behnam Madadnia, Barbara Stadlober

**Affiliations:** 1Polymer Processing, Department of Polymer Engineering and Science, Montanuniversitaet Leoben, 8700 Leoben, Austria; 2Center for Microsystems Technology, Ghent University, B-9052 Ghent, Belgium; 3Joanneum Research Forschungsgesellschaft mbH, Franz-Pichler Str. 30, 8160 Weiz, Austria; 4Designing Plastics and Composite Materials, Department of Polymer Engineering and Science, Montanuniversitaet Leoben, 8700 Leoben, Austria; 5Centre for Microsystems Technology, Imec and Ghent University, Technology Park 126, Zwijnaarde, B-9052 Ghent, Belgium

**Keywords:** injection molding, simulation, in-mold electronics, over-molding, Moldflow

## Abstract

The integration of assembled foils in injection-molded parts is a challenging step. Such assembled foils typically comprise a plastic foil on which a circuit board is printed and electronic components are mounted. Those components can detach during overmolding when high pressures and shear stresses prevail due to the injected viscous thermoplastic melt. Hence, the molding settings significantly impact such parts’ successful, damage-free manufacturing. In this paper, a virtual parameter study was performed using injection molding software in which 1206-sized components were overmolded in a plate mold using polycarbonate (PC). In addition, experimental injection molding tests of that design and shear and peel tests were made. The simulated forces increased with decreasing mold thickness and melt temperature and increasing injection speed. The calculated tangential forces in the initial stage of overmolding ranged from 1.3 N to 7.3 N, depending on the setting used. However, the experimental at room temperature-obtained shear forces at break were at least 22 N. Yet, detached components were present in most of the experimentally overmolded foils. Hence, the shear tests performed at room temperature can only provide limited information. In addition, there might be a peel-like load case during overmolding where the flexible foil might bend during overmolding.

## 1. Introduction

The need for low-cost, flexible, lightweight electronic devices has risen dramatically. Over-molding electronics is an innovative technique for producing three-dimensional (3D)-shaped smart products. The main concept is to use injection molding to create working circuit carriers with components in two-dimensional (2D) or 3D shapes. The overmolding process has been used in various applications, including the automotive sector, consumer household appliances, and medical equipment [1,2]. First, an electrical circuit is deposited on a flexible substrate, and surface mount devices (SMDs) are attached to the circuit using lead-free solder or conductive adhesives. Then, this electronic circuit is put into an injection molding machine as an insert, and the plastic flows over it. The end product is a plastic part with an integrated circuit and functional electrical components.

However, the prevailing thermal and mechanical loads during the injection molding cycle necessary to shape the molten polymer can lead to failures within the electrical circuits [3,4,5]. The mechanical forces may be considered separately as shear loads acting in-plane and pressure loads acting normally upon the component surfaces. To bypass the high stresses inherent to the injection molding process, Ott and Drummer [6] proposed using thermoplastic foam injection molding. That way, they could encapsulate epoxy-based printed circuit boards (PCBs) with cavity pressures of approx. 1 MPa only.

Heinle et al. [7] developed an injection mold that allows the forces on small components to be measured during the overmolding cycle. To that end, they added a three-way force sensor with a size of 5 × 6 × 1 mm^3^ in the center of a 40 × 40 × 2 mm^3^ mold. Hence the sensor could record the force in all directions in space: in flow (F_x_), transverse (F_y_), and perpendicular (F_z_) to the flow direction. Short shots (almost filled part and no packing pressure applied) were made, and sharp peaks of F_x_ and F_z_ were recorded during filling that decreased quickly. The centered sensor depicted no force in F_y_ because of a symmetric flow around it. During cooling, forces F_x_ and F_z_ were measured due to shrinkage.

Schirmer et al. [4,8] overmolded assembled foils with components of sizes 0402, 0603, and 0805 with polycarbonate (PC). Larger components showed lower failure rates than smaller components when the conductive adhesive was used. However, the opposite trend was observed when solder was used. Fewer crack formations were observed on components overmolded at higher injection speeds when the conductive adhesive was used, and medium injection speeds were preferred for those with solder. Further, no influence of the orientation of the components, placed in 0° (transverse flow) and 90° (in flow) orientation, was found. Components using conductive adhesive placed close to the film gate were less prone to damage. They reasoned that the melt would become significantly more viscous along the flow path, resulting in more significant damage to the components. Likewise, a faster injection speed would reduce the time the melt would cool during filling and be beneficial [4]. However, the opposite was observed for soldered resistors. Here less damage was visible on components farther away from the gate [8].

Bakr et al. [9] developed an assembled foil design comprising 18 zero-ohm resistors oriented at 0°, 45°, and 90° for overmolding in a flat or corner-shaped mold. The design was also used in this study. Two component sizes, namely, 0805 and 1206, and two copper-based foil substrates with different polymer layers, polyimide (PI) and polyethylene terephthalate (PET), were investigated. Lead-free solder and a conductive adhesive (CA) were used as interconnection materials for the PI foils and components. However, CA and a low melt temperature solder (LTS) were used for the PET foils. Only the PI foil using lead-free solder showed no change in the resistance after overmolding (in the flat mold). In contrast, the PET foils with the used interconnection materials frequently exhibited detached components. No influence of the package size, orientation, or placement within the mold was reported.

Wimmer et al. [10] overmolded assembled polymer films with glued 0805 0 ohm resistors placed in a line in a 4 mm thick test rod with PC. They estimated the shear force on the components using the injection simulation software CADmould (Simcon kunststofftechnische Software GmbH, Würselen, Germany). To that end, the part with components was modeled in the software (mesh size 0.4 mm). Three “sensors” per component with surface area A_0805_ were placed to obtain the local shear rates ($\dot{\gamma }$) and viscosities (η), and hence the shear forces F_s_ = $\dot{\gamma }$∙η∙A_0805_ at any time during the cycle [11]. Their simulation showed shear forces F_s_ = 0.8 … 1.5 N. This was substantially lower than the tested shear strength of their components of 15 N to 20 N (measured according to DIN EN 62137-1-2 at room temperature). Of the 42 films overmolded, only one component was washed away. However, components overmolded with lower melt temperatures exhibited larger increases in electrical resistance. This was denoted to crack formations due to the increased viscosity of the colder polymer [10].

State-of-the-art commercial injection molding software such as Autodesk Moldflow Insight (AMI, Autodesk Inc., San Rafael, CA, USA) [12] or Moldex3D (CoreTech System Co., Ltd., Zhubei City, Taiwan) [13] have implemented core shift analyses into their filling simulations. These features estimate possible deformation of (mold) inserts during the injection molding cycle based on the pressure distribution within the melt. Those software extensions, however, were designed with cantilevered core pins in mind that are affected by deformation when a non-uniform flow around them causes pressure differences [12].

In a very recent work [14], we investigated the integration of structural electronics in injection-molded parts. This paper used two thermoplastic polyurethanes (TPUs) as middle layers with different melting temperatures. Parameter studies were performed to investigate the influence of the melt temperature, mold temperature, injection speed, and TPU layer. A shear distortion factor for the TPU layer was derived based on simulations that linked the shear stresses with the injection time and the softening (melting) of the TPU. The distortion of the films was found to reduce with higher melt temperature, lower mold temperature, and faster injection speed. Films using the TPU with the higher melting temperature yielded significantly better results.

This literature overview shows that some studies have investigated the optimization of the injection molding process for assembled foils. However, an in-depth examination of the prevailing forces during filling is still lacking. By analyzing numerically performed parameter studies, this paper aims to gain insight into how the injection molding settings influence the forces on the components.

To that end, the foil design developed by Bakr et al. [9] was utilized (illustrated in Figure 1a), and virtual parameter studies were made using AMI. Here, the influences of the mold thickness, melt temperature, and injection speed on the forces on 1206-sized components when overmolded with PC were investigated. A Python script accessing the AMIs Synergy Application Programming Interface [15] was developed to calculate and visualize the tangential (F_t_) and normal forces (F_n_) on the components during filling (Figure 1b). The experimental results obtained in [9] were extended with additional injection molding tests, and the number of detached components after overmolding was counted (Figure 1c). In addition, the shear and peel strength values were assessed in mechanical tests (Figure 1d). The findings were then combined to extract guidelines for injection molding of assembled foils (illustrated in Figure 1e).

## 2. Materials and Methods

### 2.1. Assembled Foil Design

Figure 2a shows an assembled foil fabricated according to the design developed in [9] comprising 18 components oriented in three directions (0°, 90°, and 45°) with copper tracks used for the connection. A PET-based GTS-MP (Furukawa Electric Co., Ltd., Tokyo, Japan) foil was used with the layer structure and thicknesses given in Figure 2b.

Zero-ohm resistors of size 1206 (Yageo Corporation, New Taipei City, Taiwan) were used for the foils with dimensions presented in Table 1 [16].

Two materials for manually assembling the components on the foils’ copper tracks were used:LTS: Low-temperature Sn42Bi57Ag1 Interflux DP 5600 solder alloy (Interflux Electronics N.V., Gent, Belgium) with a melting temperature of 139 °C [17].CA: Loctite Ablestik CE 3103WLV thermoset silver-based conductive glue (Henkel AG & Co. KGaA, Düsseldorf, Germany), with a curing temperature of 120 to 150 °C [18].

### 2.2. Injection Molding

The assembled foils described above were inserted into a plate mold, as depicted in Figure 3, for overmolding with the Makrolon 2805 polycarbonate (PC) (Covestro AG, Leverkusen, Germany) [19]. The cavity thickness was set to either h = 2 mm or 3 mm. A temperature-resistant adhesive tape was used to fix the foils, and the contacts were kept free from overmolding by a retractable insert added on top of the foils (cf. gray part in Figure 3).

Table 2 lists the foils’ overmolding settings presented in [9] that were extended by the “harsh” settings of 2-260-CA and 2-260-LTS for this study. A constant injection speed of 70 cm^3^/s was used, and n = 3 foils were overmolded per setting.

A fully electric Arburg Allrounder 470 A Alldrive (Arburg GmbH + Co KG, Loßburg, Germany) injection molding machine equipped with a 25 mm screw was used. A Wittmann Tempro plus D 160 (WITTMANN Technology GmbH, Wien, Austria) temperature control unit was utilized to heat the mold to 100 °C. The dosing volume was set to either 50 cm^3^ for the 3 mm plate or 40 cm^3^ for the 2 mm plate. The switchover point (velocity-to pressure-controlled filling) was adapted for each setting to obtain an almost (99%) filled part before the start of packing. The packing pressure was set to 400 bar for 15 s, and a residual cooling time of 50 s was used for all tests.

In addition, the pressure within the cavity close to the gate was recorded using a 4 mm diameter Kistler Type 6157B (Kistler Holding AG, Winterthur, Switzerland) piezoelectric pressure sensor (see Figure 3 for sensor positioning). The retrieved pressure curves were then used to validate the simulation results.

### 2.3. Simulation

The commercial injection molding simulation software Autodesk Moldflow Insight 2021 (AMI) was used to numerically study the pressures and shear loads on the components during overmolding. To model the filling phase, AMI numerically solves the conservation equations of mass, momentum, and energy using the finite element method (FEM) [20].

The 3D FEM models were created featuring either a 2- or 3-mm plate and 18 components of size 1206. The AMI property part and part insert were assigned to the CADs of the plate and components, respectively. Thus, the AMI solver performed the flow simulation on the region of the plate only while treating the components as fixed pieces added to the cavity. The thin foil (70 µm) and the joints were not modeled. The machine die was modeled as a beam hot runner.

The part region close to the components (<2 mm) and the components were modeled with a mesh size of 0.3 mm (cf. Figure 4a). The global edge length was set to 1 mm with a minimum number of 12 elements through the thickness. The auto sizing scale factor was set to 0.9. Precise match was enabled for the contact interfaces, providing identical surface meshes for contacting bodies.

The tetrahedral element count for the 3 mm plate model was 3,198,779/65,832 (part/inserts), and for the 2 mm plate model it was 3,141,193/77,748 (part/inserts). An illustration of the 2 mm model is given in Figure 4a.

All the required simulation material data for the PC Makrolon 2805 overmolding material (such as Cross-WLF viscosity coefficients and Tait pvT coefficients) are available in the Moldflow material database (Autodesk udb-file). The components were simplified and modeled as aluminum oxide (${\mathrm{Al}}_{2}O_{3}$) ceramic. Here, the values for the density ($\rho =3.69\;g/{\mathrm{cm}}^{3}$), the specific heat capacity ($c_{p}=880\;J/(\mathrm{kg}\cdot K)$), and the thermal conductivity (λ=18.0 W/(m·K)) were taken from the literature [21].

A uniform heat transfer coefficient (HTC) of 5000 W/(m^2^∙K) (AMI default for the filling phase) was used for the part-component interfaces.

During filling, pressures (p) acting in the normal direction and shear stresses (τ) acting in the flow direction accumulated into forces in normal (F_n_) and tangential (F_t_) directions on the components, as illustrated in Figure 4b.

The direction of the tangential force (F_t_) to the center line of the mold could be specified using a shear angle (θ), as depicted in Figure 4c.

It is not feasible to receive the forces (F_n_ and F_t_ with shear angle θ) on a component (viz., part insert) during filling directly within AMI. However, among the results that AMI offers are [22,23,24]:Pressure (scalar nodal part result).Shear stress at wall (viz., frozen/molten interface; scalar nodal part surface).Velocity (vector nodal part).

A Python script that used the Synergy Application Programming Interface (API) [13] was developed to access the AMI (filling) results and information about the node location and element allocation. The mechanical forces on the inserts were estimated as described in the following and are illustrated in Figure 5:

The nodal pressure result of the part at the surface (${p_{N}}_{\mathrm{part}}$) is mapped onto the component insert mesh. This process is made straightforward by the congruent surface meshes between inserts and part (precise match mesh). Following that, a pressure average can be calculated for each insert triangle (${p_{\mathrm{Tr}}}_{\mathrm{insert}}$). By calculating the normal vector of each insert’s triangle (${{\vec{n}}_{\mathrm{Tr}}}_{\mathrm{insert}}$), pressures are operating normally upon a surface.

Similarly, the nodal shear stress result of the part (${\tau_{N}}_{\mathrm{part}}$) can be transformed into an averaged and mapped triangle result of the insert (${\tau_{\mathrm{Tr}}}_{\mathrm{insert}}$). Shear stresses act in the plane of a surface. This direction is estimated by using the closest nonzero nodal part velocity (${{\vec{v}}_{N}}_{\mathrm{part}}$) and mapping it onto the insert mesh. (The melts velocity at the surface is zero due to the assumption of wall adherence. Therefore, the velocity results are based on the closest internal part node.) An averaged vector is computed for each triangle that is projected onto the surface of the triangle and normalized (${{\vec{a}}_{\mathrm{Tr}}}_{\mathrm{insert}}$).

The area of each insert triangle is calculated (${A_{\mathrm{Tr}}}_{\mathrm{insert}}$) and multiplied by the corresponding force contribution of each triangle (${{\vec{F}}_{\mathrm{Tr}}}_{\mathrm{insert}}=\left({p_{\mathrm{Tr}}}_{\mathrm{insert}}\cdot {{\vec{n}}_{\mathrm{Tr}}}_{\mathrm{insert}}+{\tau_{\mathrm{Tr}}}_{\mathrm{insert}}\cdot {{\vec{a}}_{\mathrm{Tr}}}_{\mathrm{insert}}\right)\cdot {A_{\mathrm{Tr}}}_{\mathrm{insert}}$). The overall mechanical overmolding force of a component is assessed by adding up all the individual triangle forces (${\vec{F}}_{\mathrm{insert}}=\sum {{\vec{F}}_{\mathrm{Tr}}}_{\mathrm{insert}}$). This overall force vector can then be separated into the normal force F_n_ and tangential force F_t_ with shear angle θ.

Table 3 lists the performed simulated parameter study comprising the 8 simulations termed S01–S08. The parameter mold thickness (A) and melt temperature (B) were investigated in the experiments (listed in Table 2) at two levels. In addition, also a slower injection speed (C) was simulated.

Fill was selected as the analysis sequence, and 100 intermediate results were requested. The maximum %volume to fill per time step had to be reduced to 0.5% to ensure that this number of results was obtained. The switchover point was set to 97%, and a constant melt temperature of 100 °C was assumed for all settings.

### 2.4. Mechanical Test Setups

Shear tests on three 90°-oriented 1206 components using LTS were already performed in [9] and supplemented by equivalent tests for CA. To that end, a Bruker UMT-2 (Bruker Corporation, Billerica, MA, USA) mechanical tester platform with a purpose-built clamping plate fixture for the foils (Figure 6a) was used. The foils were firmly pressed and constrained between two metal plates, as shown in Figure 6b.

The foils were firmly pressed and constrained between a metal plate and a frame with a 6 mm wide slot, as shown in Figure 6b. Next, a shear head with a width of 4 mm was moved above the foil at 1/4th of the component’s height and connected to a 100 N load cell.

The shear tests could only be performed at room temperature at a shear head speed of 6 mm/min (following DIN EN 62137 1-2).

Peel tests were made of already overmolded components to gain insight into the peel strength of the (molded) LTS and CA connections. A 15 mm wide foil strip was laser cut for each row of 0°-, 45°-, and 90°-oriented components. An Instron 5500R (Illinois Tool Works Inc., Glenview, IL, USA) tensile test machine attached with a peel-off fixture was utilized for those tests. First, the molded PC plates were positioned with the foil strip along the centerline of the peel-off fixture. Next, the PC plates were clamped on both sides. Then, one side of the foil strip was loosened from the PC substrate to be clamped by the peel head. The pivoting fixture guaranteed a peel angle of 90°. Finally, a 10 mm/min peel speed and a 100 N static load cell—recording at a frequency of 10 1/s—were chosen. Figure 7a,b) show images of the test setup.

## 3. Results

### 3.1. Count of Detached Components

The number of detached components per overmolded foil was counted, as depicted in Figure 8. No pattern in terms of orientation or concerning gate location was noticed.

Due to the high variation of detached components within the foil molded settings, comparison across them is limited. However, a lower melt temperature (260 °C vs. 300 °C) clearly provokes detachment. Additionally, there seemed to be more detached components when a lower mold thickness (2 mm compared to 3 mm) was used and when CA was used instead of LTS to mount the components.

### 3.2. Comparison of Recorded and Simulated Injection Molding Pressures

Figure 9 contrasts the filling pressures for the 2 mm plate mold measured within the cavity using the installed pressure sensor with the simulated filling pressure curves. No packing pressure was applied for those comparisons to emphasize the pressure development during filling.

The measured and simulated pressure developments were in good agreement (similar slopes), and discrepancies in the peak pressures were likely due to differences in the application of the switchover points.

### 3.3. Simulation of Forces on Components during Filling

It is possible to investigate the force development on each component during filling through the developed Python script. Figure 10a visualizes those forces in the normal (F_n_) and the tangential directions (F_t_) on the 1206 components of setting S01. As the melt progressed, the normal force (F_n_) substantially exceeded the tangential force (Ft). Only in the initial stages of the overmolding was the tangential force (F_t_) dominant, as shown in Figure 10b. Thus, we assumed this is also the most critical moment during overmolding—an observation also mentioned in [25].

The shear angles (θ) were almost zero for the 90°-oriented components positioned close to the center line of the plate mold, as depicted in Figure 10c. For the 0°- and 45°-oriented components, however, the shear angles (θ) pointed toward the sidewalls of the mold, which was attributed to the parabolic flow front, as indicated in Figure 10d.

There seemed to be only minor differences in the shape of the force curves regarding orientation (0°, 45°, and 90°) or their position (close or far from the gate).

In Figure 10b, F_t=n_ when $\left|F_{t}\right|=\left|F_{n}\right|$ is marked with a red dot for the 90°-oriented component closest to the gate. This was done to have a clearly defined force at the beginning of the components overmolding. The F_t=n_ value is depicted for all the simulations of the parameter study in Figure 11.

Based on the calculated F_t=n_ forces stated in Figure 11, the polynomial
F_t=n_ (N) = 19.54−2.71∙A − 0.04∙B + 0.04∙C,(1)
with A as the mold thickness (mm), B as the melt temperature (°C), and C as the injection speed (cm^3^/s) can be derived (coefficient of determination R^2^ = 97%). The related effects plot is shown in Figure 12. In such diagrams, the examined factor is varied between the two levels, while the other factors are set to the intermediate value in the regression equation (Equation (1)). Accordingly, as the force F_t=n_ becomes lower, the thicker the cavity (A), the higher the melt temperature (B), and the slower the injection speed (C).

In the simulations, one can easily set the (filling) pressures to zero (${p_{N}}_{\mathrm{part}}=0$), so that only the shear stresses (${\tau_{N}}_{\mathrm{part}}$) remain. Figure 13a shows that in such a hypothetical case, the calculated tangential forces (F_t_*) would become significantly lower compared to the previously calculated tangential force (F_t_). For instance, the previously defined force F_t=n_ would be halved ($F_{t=n}=F_{t}≅2\cdot {F_{t}}^{*}$). Hence there is a substantial contribution of the pressure gradient during filling, despite the small size of the components (Figure 13b).

### 3.4. Mechanical Test Evaluation

Figure 14 shows that roughly twice as high shear loads at break can be obtained when LTS (42.8 ± 2.7 N, [9]) is used to mount the 1206 components to the copper pads of the foils compared to when CA (21.9 ± 3.2 N) is used.

Figure 15 shows the peel load vs. peel length plots of the cut 15 mm wide foil strips containing the already overmolded 1206 components, as described in Section 2.4 above.

Figure 15a shows a significant drop in the peel load whenever a component mounted using CA is reached. This was not apparent for LTS-mounted components (Figure 15b), where an initial increase in the peel curve is frequently observed. All peeled components remained in the PC substrate (Figure 15c). Compared to CA, LTS components can be exposed to higher load cases that resemble peel loads. Load cases like this might occur during overmolding when the flexible foil potentially bends during overmolding, as illustrated in Figure 15d.

## 4. Conclusions

The injection molding simulations—in which the force development on 1206-sized components during overmolding with PC in a plate mold was investigated—may be summarized as follows: The tangential forces F_t_ (acting in the “flow direction”) on the components are only dominant in the initial stage of the overmolding. They are quickly surpassed in magnitude by the normal forces F_n_ (pressing the components to the mold wall). Hence, the potential detachment of components is expected to happen in the initial moments of the overmolding. This observation was also suggested in [25]. The force F_t=n_, when $\left|F_{t}\right|=\left|F_{n}\right|$, seems to be almost independent of the component’s orientation (0°, 45°, and 90°) or positioning within the mold (close to or far from the gate). The calculated shear angle θ indicated a push in the direction of the mold sidewalls for components placed at a distance to the center line of the mold caused by the parabolic polymer flow.

According to the simulated parameter study, the force on the components can be reduced by increasing the mold thickness, using a higher melt temperature, and using a slower injection speed. The calculated F_t=n_ ranged from 1.34 N (S03) to 7.23 N (S06) for the 1206 components. Those simulated tangential forces are substantially lower than the experimentally obtained shear loads at breaks of ~22 N and ~43 N for the conductive adhesive (CA, Loctite Ablestik CE 3103WLV) and low-temperature solder (LTS, Interflux DP 5600), respectively. This is a discrepancy to the observation that most of the overmolded assembled foils have detached components. A possible explanation could be that the shear testing was performed at room temperature (following DIN EN 62137 1-2), while injection molding was conducted at elevated temperatures. Further, the flexible foil might cause a more peel-like load case during overmolding. Peel tests of foil strips with overmolded components indicated higher peel loads when LTS was used instead of CA.

The assembled foils overmolded in the 2 mm plate mold at low melt temperature (260 °C) showed more detached components compared to the assembled foils overmolded in the 3 mm plate mold at high melt temperatures (300 °C). This is an expected trend compared to the simulated forces (F_t=n_). Seemingly, when a higher melt temperature is used, the viscosity is lower, and thus lower forces on the components outweigh the potential faster softening of the connections.

Further, no pattern for the detachments could be observed regarding the components’ orientations (0°, 45°, or 90°) or distance from the gate. This matches the simulation results, which, as discussed, also do not show substantial differences in the calculated forces (regarding F_t=n_).

The pressure gradients’ contribution to the overall component’s tangential force $F_{t}$ is considerable. An examination of the shear stresses alone (as done in [10]) would therefore result in a significant underestimation of the prevailing component’s tangential forces ($F_{t=n}=F_{t}≅2\cdot {F_{t}}^{*}$).

## Figures and Tables

**Figure 1 micromachines-14-00876-f001:**
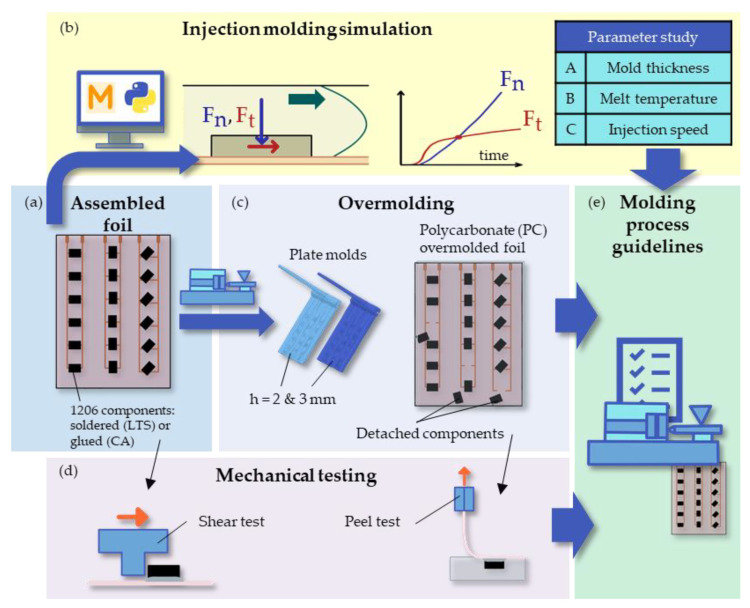
Workflow to study the impact of injection molding parameters on detached components: The foil design with soldered resistor components developed in [9] (**a**) was numerically simulated in parameter studies in which the normal (F_n_) and the tangential forces (F_t_) on the components during filling were investigated (**b**). Additional injection molding tests were conducted on the results in [9], and the detached components were counted after overmolding (**c**). Further mechanical tests like shear and peel tests investigating the mountings’ strength were made (**d**). Finally, the findings concerning simulation, overmolding process, and mechanical testing were combined to extract guidelines for injection molding of assembled foils (**e**).

**Figure 2 micromachines-14-00876-f002:**
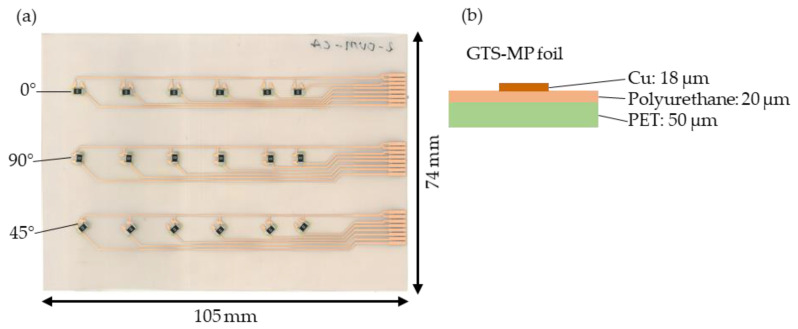
Assembled foil for overmolding (developed in [9]) using 1206 resistor components (**a**) and a layer structure of the foil (**b**).

**Figure 3 micromachines-14-00876-f003:**
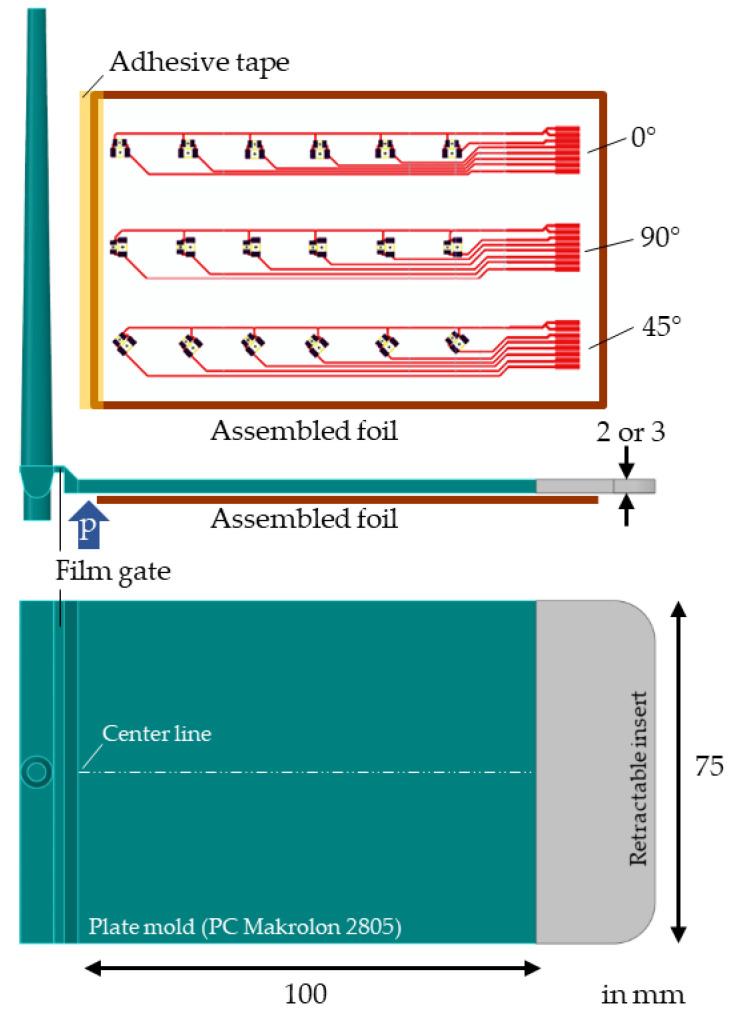
Setup of the plate mold with dimensions, where the foils were placed on the bottom side for overmolding. The insert, drawn in grey, was used to keep the foil contacts free from overmolding. As the blue arrow indicates, the mold was equipped with a pressure sensor close to the gate.

**Figure 4 micromachines-14-00876-f004:**
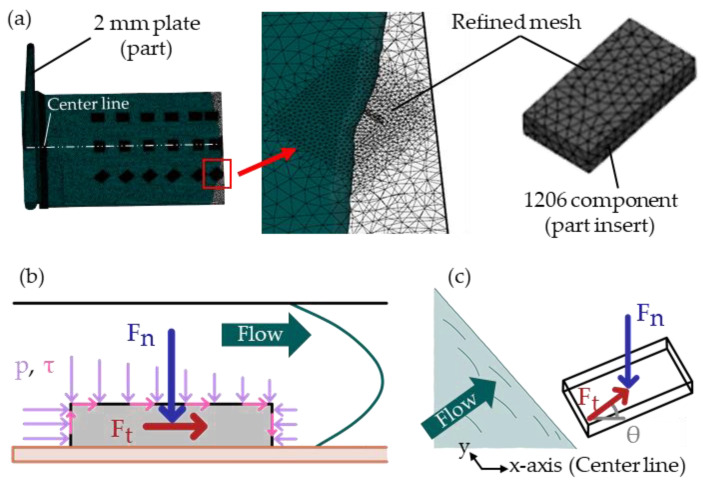
Filling illustration of the 2 mm plate AMI model (**a**). Pressures (p) and shear stresses (τ) accumulate into normal (F_n_) and tangential (F_t_) forces on the components during filling (**b**). A shear angle (θ) might be used to specify the direction of the tangential force (F_t_) with respect to the center line (**c**).

**Figure 5 micromachines-14-00876-f005:**
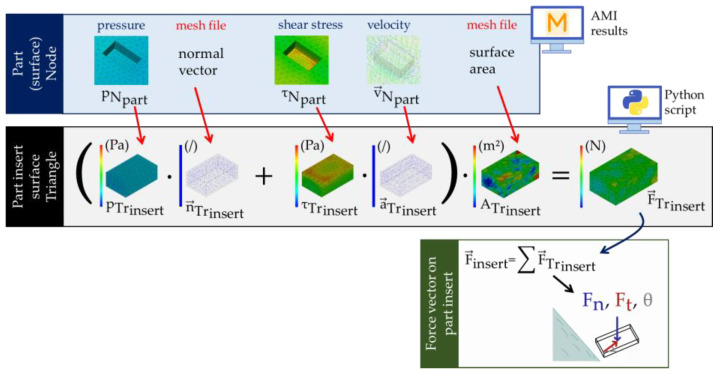
Illustration of the Python API script’s functional principle.

**Figure 6 micromachines-14-00876-f006:**
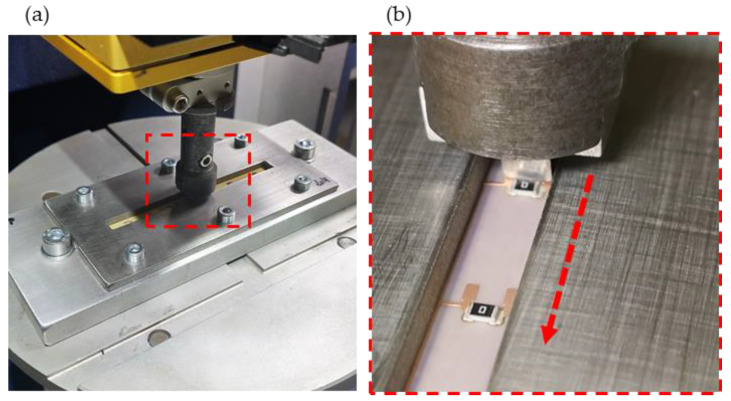
The foils were fixed within a clamping plate fixture (**a**) and sheared off (**b**).

**Figure 7 micromachines-14-00876-f007:**
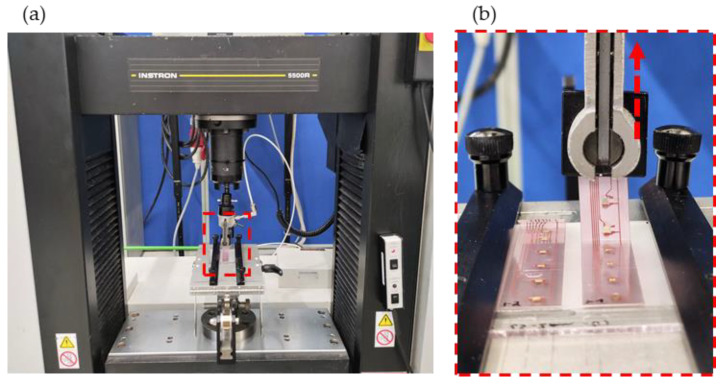
Strips of already overmolded foils with components were laser cut and fixed in a peel-off fixture (**a**) and peeled at an angle of 90° (**b**).

**Figure 8 micromachines-14-00876-f008:**
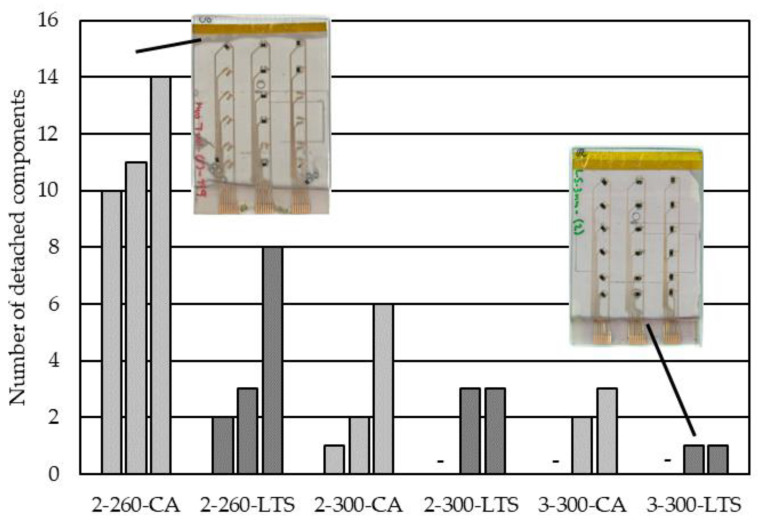
The number of detached components per setting (each foil comprising 18 components total). The labeling reads mold thickness (mm)–melt temperature (°C)–material for assembling (/).

**Figure 9 micromachines-14-00876-f009:**
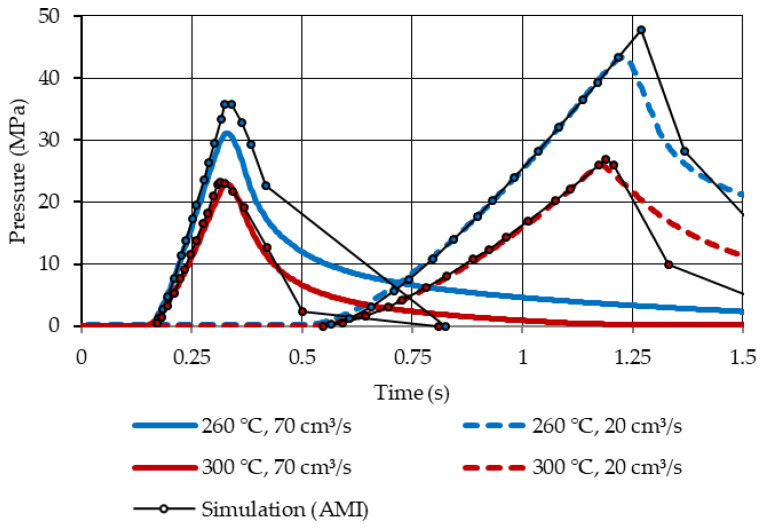
Comparison of the recorded (in blue and red) and simulated (in black) pressure curves for short shots (no packing pressure applied) with the 2 mm plate mold at different melt temperatures and injection speeds. The positioning of the pressure sensor is marked with a blue arrow in Figure 3 (the mold temperature was set to 100 °C).

**Figure 10 micromachines-14-00876-f010:**
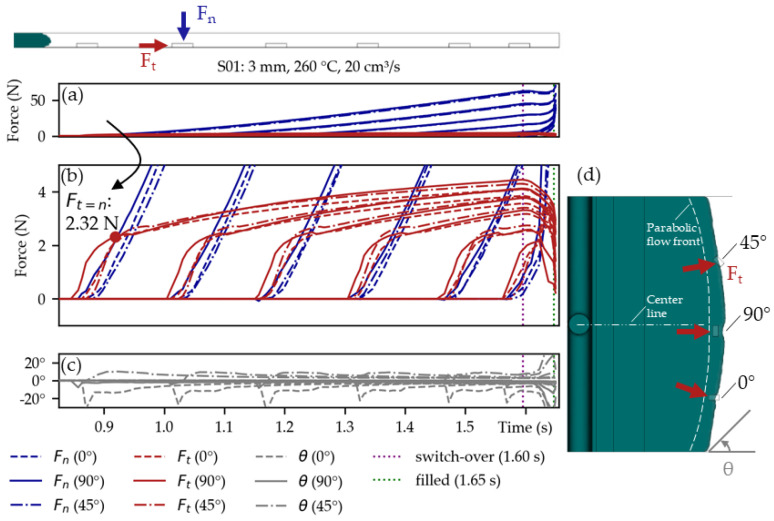
Simulated normal (F_n_) and tangential (F_t_) forces on the 1206 components during overmolding in (**a**) with augmentation on the tangential forces in (**b**). The 90°-oriented components along the center line show a shear angle (θ) close to zero (**c**). However, the 0°- and 45°-oriented components yield shear angles (θ) pointing toward the mold side walls, which can be attributed to the parabolic flow front (**d**).

**Figure 11 micromachines-14-00876-f011:**
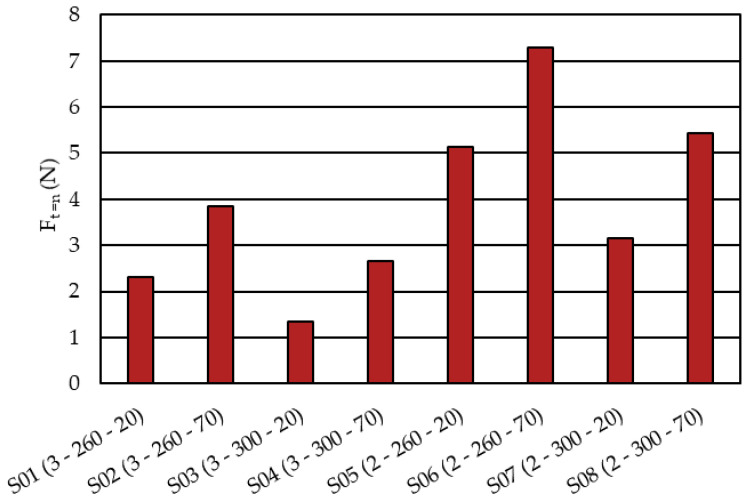
Simulated F_t=n_ forces for the parameter study defined in Table 3. The numbering in brackets reads (mold thickness (mm)–melt temperature (°C)–injection speed (cm^3^/s)).

**Figure 12 micromachines-14-00876-f012:**
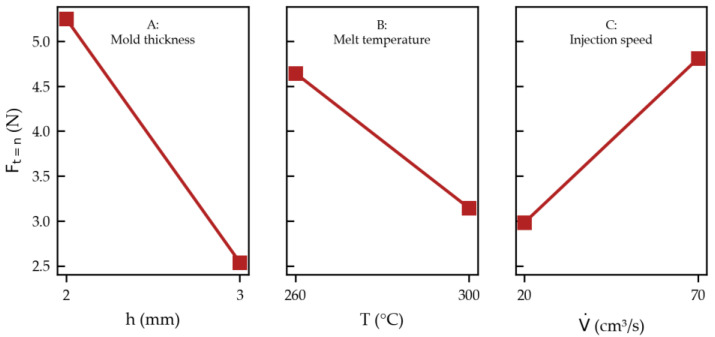
Effect plot for F_t=n_ according to Equation (1).

**Figure 13 micromachines-14-00876-f013:**
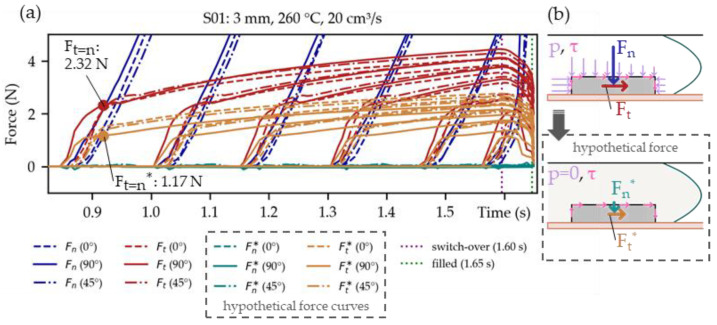
Comparison of the simulated normal (F_n_) and tangential (F_t_) forces with the hypothetical force curves in the normal (F_n_*) and tangential (F_t_*) direction (**a**). For the latter, the filling pressures were virtually set to zero (${p_{N}}_{\mathrm{part}}=0$) as illustrated in (**b**). Therefore, the calculated tangential forces (F_t_*) were lower compared to the tangential forces (F_t_) obtained previously (${F_{t=n}}^{*}≅50\%\cdot F_{t=n}$ ), indicating the substantial contribution of the pressure gradient during filling.

**Figure 14 micromachines-14-00876-f014:**
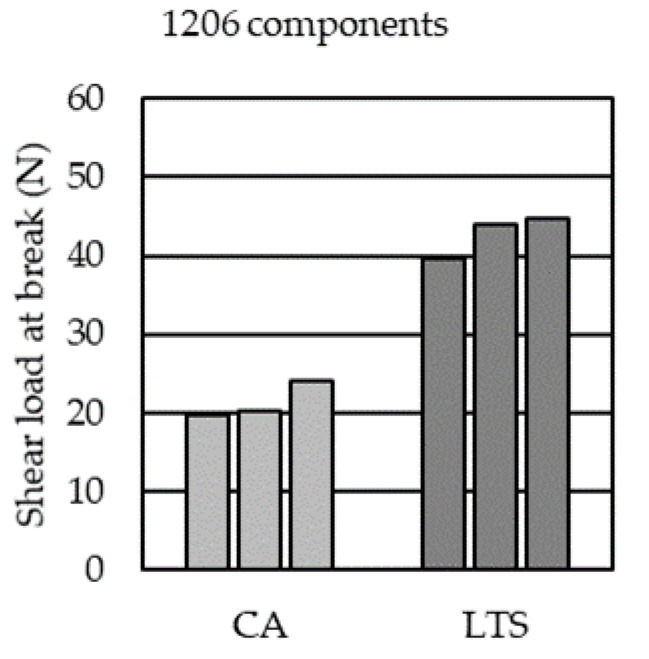
Shear load at break for 1206 components using CA or LTS to mount the copper pads.

**Figure 15 micromachines-14-00876-f015:**
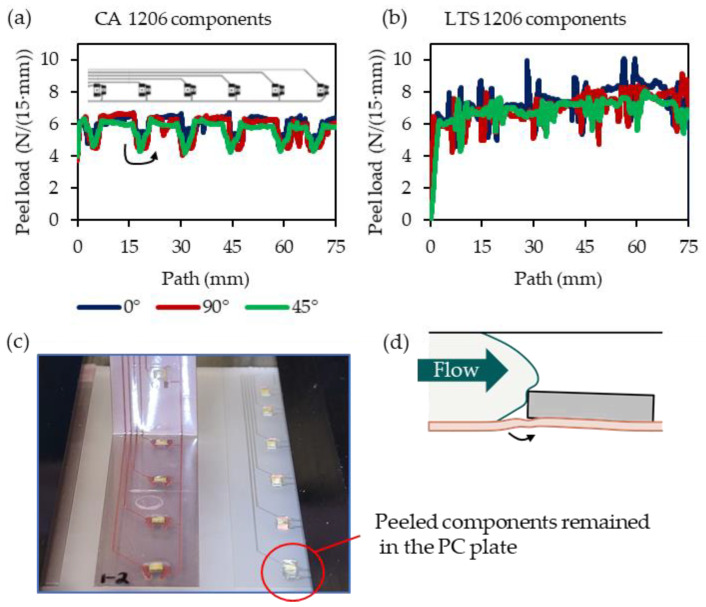
Peel force of the film strips cut from overmolded foils comprising components using low conductive adhesive (**a**) and low-temperature solder (**b**). All peeled components remained in the PC substrate (**c**). Similar load cases might occur during overmolding when the flexible foil potentially bends during overmolding (**d**).

**Table 1 micromachines-14-00876-t001:** Dimensions of the components used [16].

Type	Length (mm)	Width (mm)	Height (mm)
1206	3.10 ± 0.10	1.60 ± 0.10	0.55 ± 0.10

**Table 2 micromachines-14-00876-t002:** The molding plane of the foils produced in [9] was extended by setting 2-260-CA and 2-260-LTS (mold temperature: 100 °C, injection speed: 70 cm^3^/s, n = 3).

Setting No.	Mold Thickness (mm)	Material of Assembly	Melt Temperature (°C)
2-260-CA	2	CA	260
2-260-LTS	2	LTS	260
2-300-CA	2	CA	300
2-300-LTS	2	LTS	300
3-300-CA	3	CA	300
3-300-LTS	3	LTS	300

**Table 3 micromachines-14-00876-t003:** Performed injection molding simulations.

Setting No.	A—Mold Thickness (mm)	B—Melt Temperature (°C)	C—Injection Speed (cm^3^/s)
S01	3	260	20
S02	3	260	70
S03	3	300	20
S04	3	300	70
S05	2	260	20
S06	2	260	70
S07	2	300	20
S08	2	300	70

## Data Availability

Not applicable.

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
