# Peer review of "Parameter Study on Force Curves of Assembled Electronic Components on Foils during Injection Overmolding Using Simulation"

_micromachines, 2023, doi:10.3390/mi14040876_

Round 1

Reviewer 1 Report

This manuscript tried to reveal the insight about how the injection molding conditions influence the forces on the components by simulation. The comments and questions are:

1.        Line 205, 3,198,779 / 65,832 205(part / inserts). What is the meaning of part/inserts? Please give the element number about the model.

2.        Figure 4, please make figure(a) clearer. It is hard to see the detail mesh.

3.        Figure 5, make the figure larger and clear.

4.        What is the meaning of “S01”? Sample no. 1? Or setting no. 1? It should be explained in the main content.

5.        Ft=n is function about thickness, melt temperature and injection speed. There three subfigures in Figure 12, please give the conditions for each subfigures. For example, first figure is about mold thickness, what are the values of melt temperature and injection speed?

6.        Why the authors used constant injection speed in table 2 while used two different injection speed in table 3? It should be explained and kept consistent.

Author Response

Dear reviewer!

Thank you very much for your exact and specific comments! We are trying to address them one by one in the following:

Comment 1:
"3,198,779" is the number of tetrahedra with the property "part" (viz. the 3 mm plate) assigned, and "65,832 205" is the number of tetrahedra with the property "part insert" (viz. the 16 components). This distinction must be made in the simulation software (AM). Therefore, we added the following clarification in line 198: "The AMI property part and part insert were assigned to the CADs of the plate and components, respectively. Thus the AMI solver performs the flow simulation on the region of the plate only while treating the components as fixed pieces added to the cavity."

Comment 2:
We agree that it is hard to see the mesh in detail in Figure 4 (a). We tried to fix this by rearranging and enlarging the corresponding subfigures. In general, we could not make a FEM mesh representation (in AMI) of our model that fully satisfied us.

Comment 3:
We had a second look at Figure 5, which was indeed convoluted. We tried to improve the illustration by rearranging the individual images, and we added some descriptions. Hopefully, the figure is now easier to grasp and more meaningful.

Comment 4:
Indeed, this needs to be covered in the main text. Thank you for noticing! We added the following clarification to line 261: "comprising the 8 simulations termed S01-S08".

Comment 5:
We skipped explaining the concept of the effects plot. We have now made up for this in line 353: "In such diagrams, the examined factor is varied between the two levels, while the other factors are set to the intermediate value in the regression equation (Eq. (1))."

Comment 6:
Due to limited available material, we kept the injection speed constant (70 ccm/s) in the experiments. Thus we could experimentally investigate two materials for assembly (LTS and CA) and still repeat the individual experiments (n=3; estimation of variability). However, we (obviously) did not face this restriction in the simulations and could add the second slower injection speed (20 ccm/s). We mentioned this in lines 262-264: "The parameters mold thickness (A) and melt temperature (B) were investigated in the experiments (listed in Table 2) at two levels. In addition, also a slower injection speed (C) was simulated."

Thank you again for your clear remarks. We hope we have addressed it sufficiently.

With Best Regards.

Reviewer 2 Report

In their manuscript:"parameters study on force cures of assembled electronic components on foils during injection overmolding using simulation", Hubmann, et al. use simulations to analyze force and stress distributed in the material during the overmolding process. The method and presentation are scientifically sound and I recommend publication.

I only have a couple of questions from the authors:

1.  As I understand you use all your data points from figure 11 to calibrate equation 1. Would you be able to cross examine equation 1?

2. Figure 5 is very difficult to understantd. would you be able to rearrange it?

Author Response

Dear reviewer!

Thank you very much for your exact and specific comments! We are trying to address them one by one in the following:

Comment 1:
Indeed, all the simulation data (S01-S08) was used to derive Eq. (1). Inherent to 2-factorial designs is the assumption of linear relationships of the factors (A, B, and C). The quality of the linear interpolation when using Eq. (1) was not checked. However, each factor's effect (size and direction when changing from the low to the high level) could be determined.

Comment 2:
We had a second look at Figure 5, which was indeed convoluted. We tried to improve the illustration by rearranging the individual images, and we added some descriptions. Hopefully, the figure is now easier to grasp and more meaningful.

Thank you again for your clear remarks. We hope we have addressed it sufficiently.

With Best Regards.

Round 2

Reviewer 1 Report

The manuscript has been revised accordingly.